# Distribution of Cleaved SNAP-25 in the Rat Brain, following Unilateral Injection of Botulinum Neurotoxin-A into the Striatum

**DOI:** 10.3390/ijms24021685

**Published:** 2023-01-14

**Authors:** Friederike Schümann, Oliver Schmitt, Andreas Wree, Alexander Hawlitschka

**Affiliations:** 1Institute of Anatomy, Rostock University Medical Center, Gertrudenstraße 9, 18057 Rostock, Germany; 2Medical School Hamburg, Am Kaiserkai 1, 20457 Hamburg, Germany

**Keywords:** Botulinum neurotoxin-A, SNAP-25, striatum, densitometry, connectome, rat brain, acetylcholine, Parkinson’s disease, basal ganglia, axonal transport

## Abstract

In Parkinson’s disease, hypercholinism in the striatum occurs, with the consequence of disturbed motor functions. Direct application of Botulinum neurotoxin-A in the striatum of hemi-Parkinsonian rats might be a promising anticholinergic therapeutic option. Here, we aimed to determine the spread of intrastriatally injected BoNT-A in the brain as well as the duration of its action based on the distribution of cleaved SNAP-25. Rats were injected with 1 ng of BoNT-A into the right striatum and the brains were examined at different times up to one year after treatment. In brain sections immunohistochemically stained for BoNT-A, cleaved SNAP-25 area-specific densitometric analyses were performed. Increased immunoreactivity for cleaved SNAP-25 was found in brain regions other than the unilaterally injected striatum. Most cleaved SNAP-25-ir was found in widespread areas ipsilateral to the BoNT-A injection, in some regions, however, immunoreactivity was also measured in the contralateral hemisphere. There was a linear relationship between the distance of a special area from the injected striatum and the time until its maximum averaged immunoreactivity was reached. Moreover, we observed a positive relationship for the area-specific distance from the injected striatum and its maximum immunoreactivity as well as for the connection density with the striatum and its maximum immunoreactivity. The results speak for a bidirectional axonal transport of BoNT-A after its application into the striatum to its widespread connected parts of the brain. Even one year after BoNT-A injection, cleaved SNAP-25 could still be detected.

## 1. Introduction

In this work, we systematically analyzed for the first time how BoNT-A distributes within the rat brain after a unilateral injection into the striatum (CPu), considering the temporal dynamics within one year. This study was performed because in previous work, we performed intrastriatal BoNT-A injections in a unilateral Parkinson animal model, but the specific distribution of BoNT-A after such treatment remained unknown.

Parkinson’s disease (PD) is the most frequent neurodegenerative movement disorder that is clearly related to advanced age [1,2,3,4]. In the course of the disease, there is a loss of dopaminergic neurons in the Substantia nigra pars compacta (SNpc), whose axons project into the striatum of humans (= Caudate-Putamen complex = CPu in rats). In the CPu, dopaminergic afferents from the SNpc inhibit a small but important contingent of spontaneously active cholinergic interneurons. If these cholinergic interneurons are not inhibited due to decreased dopamine in PD, they locally release an excess of acetylcholine. This hypercholinism is partly responsible for a large part of the hypokinetic symptoms of PD [5]. Since the 1960s, Parkinson’s disease has been treated preferably with L-3,4-dihydroxyphenylalanine (L-DOPA) [6,7,8]. After an initial “honeymoon phase” with good anti-Parkinsonian effects, the effectiveness against Parkinson’s symptoms decreases with increasing duration of treatment and other side effects such as dyskinesia and dystonia come to the fore. Even at the beginning of the treatment with L-DOPA, the treatment is accompanied by nausea and disturbances of impulse control [8,9]. To reduce the premature degradation of L-DOPA in the body periphery and thus increase the amount of available L-DOPA entering the CNS through the blood–brain barrier, catechol-O-methyltransferase inhibitors such as entacapone and tolcapone and DOPA decarboxylase inhibitors such as carbidopa and benserazide are additionally administered [7,10]. To slow down the breakdown of dopamine in the CNS, treatment is performed with blood–brain barrier-crossing monoamine oxidase B inhibitors such as rasagiline and selegiline [11,12,13].

Prior to the establishment of L-DOPA therapy, treatment with centrally acting anticholinergics was the method of choice for the treatment of PD, and anticholinergics remain a mainstay of PD therapy today [10,14,15,16,17,18]. By administering blood–brain barrier-penetrating anticholinergics, the aim is to counteract striatal hypercholinism in PD, which leads to increased activation of medium spiny neurons embedded in the indirect pathway of basal ganglia loops. One of the best-known preparations is biperiden (Akineton^®^). Therapy with anticholinergics has good to very good anti-Parkinsonoid effects, but since the respective preparations are applied systemically, their use is accompanied by a large number of peripheral and central nervous system side effects. These include mydriasis, accommodation problems, increase in intraocular pressure, dry mouth, inflammation of the salivary glands, dry eyes, muscle pain, loss of strength, changes in the voice, dysphagia, decreased peristalsis of the esophagus, and associated regurgitation, constipation, urinary retention, prostate problems, tachycardia, fever in warm/higher ambient temperature, fatigue, dizziness, hallucinations, memory impairment and confusion. In particular, the cognitive disturbances that often accompany PD can be exacerbated by therapy with anticholinergics because, in contrast to CPu, hypocholinism is more likely to occur in the basal nucleus of Meynert in PD. Gait disturbances may also worsen, as acetylcholine deficiency also occurs in the pedunculopontine nucleus in PD [14,19,20,21,22,23,24,25,26,27,28]. Therefore, anticholinergic interventions with effects locally limited to the CPu would be desirable to counteract striatal hypercholinism and avoid adverse anticholinergic effects in other brain regions and/or in the periphery.

For a decade, there has been an experimental approach to treat rat’s hemi-Parkinsonism-associated striatal hypercholinism by intrastriatal application of Botulinum neurotoxin-A (BoNT-A) to improve motor symptoms [29,30,31,32,33]. BoNT-A binds to the SV2 receptor of the presynaptic membrane and is then taken up into the presynapse by endocytosis. After the light chain of BoNT-A has left the lysosome, it cleaves the Synaptosomal-Associated Protein-25 kD (SNAP-25), an important component of the N-ethylmaleimide-sensitive-factor attachment receptor (SNARE) complex. This blocks the fusion of transmitter vesicles with the presynaptic membrane, preventing the release of acetylcholine for months [34,35,36,37,38,39,40].

First results have been obtained by unilateral striatal BoNT-A injection in the 6-hydroxydopamine (6-OHDA)-induced rat model of hemi-Parkinsonism. In the 6-OHDA-induced hemi-Parkinsonian model, stereotactic injection of 6-hydroxydopamine into the medial forebrain bundle leads to the death of dopaminergic neurons of the SNpc and their efferent axons into the CPu. The therapeutic potential of intrastriatal BoNT-A application for PD was initially investigated in the past in the 6-OHDA-induced rat hemi-Parkinson model, as this has been a well-established and valid animal model for decades, offering a number of advantages. The basal ganglia loops are disturbed by the unilateral lesion on only one side of the brain. The healthy side serves as a quasi-reference and the imbalance between the healthy and the lesioned hemisphere in terms of motor control of the two body halves makes some motor behavior tests possible in the first place, to measure even discrete disturbances or improvements in motor control in the course of experimental treatments [41,42,43,44,45,46]. For example, only in the hemi-Parkinson model the apomorphine-induced rotation test, the amphetamine-induced rotation test, the cylinder test, and the corridor task are possible. The apomorphine-induced rotation test is based on a compensatory upregulation of the D_2_ receptor concentration in the lesioned CPu, whereby the animals rotate against the lesion side depending on the lesion success after systemic administration of apomorphine. The amphetamine-induced rotation test is based on the inability of dopamine release in the lesioned CPu due to the loss of dopaminergic afferents from the SNpc. Animals rotate ipsilaterally to the lesion side in the amphetamine-induced rotation test after systemic administration of amphetamine [47,48,49,50,51,52]. In the cylinder test, the ratio of the use of both front paws during spontaneous raising and bracing of the animals and thus the spontaneous use of the left and right front paw is determined. In hemi-Parkinsonian animals, the paw contralateral to the lesion is used significantly less, sometimes not at all, for bracing [53,54]. In the corridor task, hemi-Parkinsonian animals are examined for a neglect. Hemi-Parkinson rats show a clear neglect of their surroundings, which are located contralateral to the lesion [55].

Dopaminergic afferents also project to the D_2_ receptor bearing tonic active cholinergic interneurons and inhibit them under physiological conditions. The cholinergic interneurons in turn project to GABAergic projection neurons, which inhibit the lateral pallidum segment. We hypothesize that much of the motor dysfunction in both human PD and the measurable deficits in the rat hemi-Parkinson model is due to a lack of inhibition of cholinergic interneurons in the CPu by dopamine. This lack of inhibition due to a loss of dopaminergic afferents leads to an increased release of acetylcholine in the dopaminergically deafferented CPu of the 6-OHDA-hemilesioned rat. Our working hypothesis is that blocking the excessive acetylcholine release in the CPu should improve motor symptoms. It is well-known that BoNT-A inhibits the release of acetylcholine. We measured improvements in the mentioned motor tests after intrastriatal BoNT-A application to 6-OHDA hemilesioned rats in previous studies [31,32,33,56,57]. These tests also revealed that the BoNT-A effect lasts approximately three months. Further experiments in which BoNT-A was repeatedly injected intrastriatally indicated that BoNT-A exerted a persistent effect in the brain of the tested rats beyond three months [31]. In the course of our studies, we want to investigate whether the intrastriatal application of BoNT-A is able to improve motor impairments in the 6-OHDA hemi-Parkinson rat model, by its anticholinergic character. The aim is to avoid the peripheral and central side effects of previous anticholinergic therapies, which are based on systemic application, by injecting only into the hypercholinism-affected CPu.

In the past, there have been many reports that BoNT-A is also actively transported away from its injection site, both anterogradely and retrogradely [58,59,60,61,62]. To determine in advance whether this is also the case when injected into the CPu and whether this is in keeping with the claim to prevent side effects in other brain areas, in this work over the course of a year, we investigated whether and to what extent BoNT-A is distributed in the rat brain after intrastriatal injection. Immunohistochemical labeling and densitometric measurement of the cleavage product of BoNT-A, the cleaved SNAP-25, was performed in brain sections of rats treated unilaterally with BoNT-A. Up to one year post-injection survival, treated and untreated hemispheres were compared. Our aim was to test the intensity and duration of BoNT-A catalytic activity in CPu and neuronally connected areas after a single BoNT-A application (Figure 1). In parallel, sham-treated control animals were studied. Furthermore, we wanted to answer the following questions:

(i) Does BoNT-A spread into other parts of the brain after injection into the CPu, and if so, does this spread tend to be in the form of nondirectional diffusion or does evidence for axonal transport corroborate?

(ii) To which nuclei and brain areas is BoNT-A transported after an intrastriatal injection?

(iii) Is there a time-dependent order of the occurrence of BoNT-A or cleaved SNAP-25 in these different brain areas?

## 2. Results

### 2.1. Verification of Successful BoNT-A Injection—Botulinum Neurotoxin-A-Induced Varicosities

The success of BoNT-A application was demonstrated by detecting tyrosine hydroxylase (TH)-positive and choline acetylcholine transferase (ChAT)-positive swellings on nerve fibers in the treated CPu in immunohistochemical staining. These swellings had a diameter of about 1 to 5 µm, previously referred to as BoNT-A-induced varicosities (BiVs) (Figure 2 and Figure 3). We previously demonstrated that these swellings are always detectable following successful BoNT-A application, but were never found in sham-treated brains [31,33,63].

### 2.2. General Findings and Qualitative Evaluation

Already, macroscopically it could be seen that the intensity of the immunohistochemical staining against cleaved SNAP-25 was different in various areas. The immunoreactivity did not decrease gradually from the injection site but was more pronounced in some more distant areas, such as the SN, than in some areas closer to the injection site. The strongest immunohistochemical staining for cleaved SNAP-25 was measured at each time point in the SN. Fibers were always immunoreactive for cleaved SNAP-25. In the olfactory bulb, however, cell bodies were additionally stained. Throughout the BoNT-A-injected brains, no degenerative changes were found.

#### Olfactory Bulb

The olfactory bulb (OB) was the only brain region in which neuronal cell bodies, not just fibers, were immunoreactive for cleaved SNAP-25 after intrastriatal BoNT-A injection. Increased immunoreactivity was observed throughout the OB, especially the OB ipsilateral to the BoNT-A injection. We observed nerve cells labeled for cleaved SNAP-25, especially in the granule cell and the mitral cell layers. Labeled fibers were found throughout the OB, most concentrated in the internal and external plexiform layers (Figure 4). The maximum immunoreactivity against cleaved SNAP-25 was observed after three months, but immunoreactive neurons and fibers were also found as early as two weeks and up to nine months after treatment. After 12 months, immunoreactive perikarya were found only sporadically. Nevertheless, some cleaved SNAP-25-positive fibers and neurons were also found in the contralateral OB up to six months after BoNT-A treatment, whereas no positive fibers or cell bodies were found in OBs from sham-treated rats at any time.

### 2.3. Comparison of the Optical Densities of Investigated Brain Areas between Treated and Untreated Hemisphere

Table 1 lists the mean optical densities (OD) of all examined structures at each examination time point. Structures marked “ips” are located ipsilateral, while those marked with “con” are located contralateral to the treated CPu (Table 1).

#### 2.3.1. Caudate-Putamen Complex (CPu)

High immunoreactivity against cleaved SNAP-25 was observed in the right BoNT-A-treated CPu, whereas we observed no or low immunoreactivity in the contralateral CPu. The OD of the BoNT-A-treated CPu are significantly higher than those of the contralateral, untreated CPu at all time points (Figure 5, Table 1).

#### 2.3.2. Globus Pallidus (GP)

Additionally, the OD of the GP on the BoNT-A-treated side were significantly higher than those of the contralateral, untreated hemisphere (Figure 5, Table 1).

#### 2.3.3. Entopeduncular Nucleus (EP)

The OD of the EP nucleus on the treated side were significantly higher than those of the contralateral, untreated side (Figure 5, Table 1).

#### 2.3.4. Substantia Nigra (SN)

The right SN (ipsilateral to the injection side) of the BoNT-A-treated animals was the nucleus with the highest OD of all measured structures (Table 1). The OD of the SN on the treated side were significantly higher at all time points than those of the contralateral, untreated side (Figure 5, Table 1). Nevertheless, a clear staining could also be seen macroscopically in the contralateral SN.

#### 2.3.5. Medial Thalamic Nuclei (MThal)

In the MThal on the treated side as well as on the untreated side, an increased OD could be detected by qualitative observation. Only in the group finalized two weeks after treatment was a significantly higher OD found on the treated side than on the contralateral side (Figure 5, Table 1).

#### 2.3.6. Ventral Thalamic Nuclei (VThal)

Two weeks and three months after treatment, the OD of the VThal on the treated side was significantly higher than on the untreated side. Interestingly, the nucleus reticularis clearly stood out from the rest of the VThal. Similarly, the nucleus reticularis was also stained on the contralateral hemisphere and stood out even more clearly from the macroscopically unstained or only lightly stained remaining VThal (Figure 5, Table 1).

#### 2.3.7. Motor Cortex (MC)

The MC of the treated side showed a significantly higher OD 1 and 12 months after treatment on the treated side (Figure 5, Table 1).

#### 2.3.8. Accumbens Nucleus (Acb)

The OD of the Acb of unilaterally BoNT-A-treated animals was significantly higher on the treated side than on the contralateral side at almost all measurement time points. Only 9 months after treatment these differences were not significant (Figure 5, Table 1).

#### 2.3.9. Pons (Pn)

The pontine nuclei on both sides showed distinct immunoreactivity to cleaved SNAP-25. There were no significant differences in the OD of the Pn of the BoNT-A-treated and the untreated sides at all time points (Figure 5, Table 1).

#### 2.3.10. Ventral Tegmental Area (VTA)

The VTA showed only slight immunoreactivity to cleaved SNAP-25. The differences in OD between the left and right side were only significant at the measurement point at month 3 after treatment (Figure 5, Table 1).

#### 2.3.11. Habenular Nuclei (Hb)

The Hb also showed distinct immunoreactivity to cleaved-SNAP-25, but no significant differences were measured between the treated and untreated sides (Figure 5, Table 1).

#### 2.3.12. Piriform Cortex (Pir)

The Pir showed distinct immunoreactivity to cleaved-SNAP-25. The OD was significantly more pronounced on the treated side than on the contralateral side. Only at month 9 after treatment were the differences between the treated and untreated sides not significant (Figure 5, Table 1).

#### 2.3.13. Basolateral Amygdala Nuclei (BLAm)

The OD of BLAm nuclei on the treated side was always significantly higher than on the contralateral side except for at 6 and 9 months after treatment (Figure 5, Table 1).

### 2.4. Temporal Dynamics of the Optical Densities (OD) of the Investigated Brain Parts

The calculated OD were graphically displayed in their temporal progression over time by the “cubic-spine interpolation” function in the program Origin^®^. In the process of the graphical evaluation, it was noticeable that most of the OD of the nuclei and brain regions had very similar dynamics. After an initial decrease and the reaching of a minimum, the OD increased again until a maximum was reached within the following months. Afterwards, the OD decreased continuously until the end point (Figure 6). The minima and maxima of the optical densities of the investigated structures determined by cubic-spine extrapolation were analyzed with respect to their mean value and the time of their occurrence. In addition, the minimum and maximum values obtained during each measurement were also extrapolated and graphically plotted. The data for maxima and minima determined in this way are not actually measured values, but represent virtual, extrapolated values. Nevertheless, time windows for reaching the extrema in the immunoreactivity for cleaved SNAP-25 in the different brain areas can be determined.

For evaluation of the occurrence of temporary maxima and minima of immunoreactivity for cleaved SNAP-25 in each brain area, three curves were plotted on each graph. One curve includes the lowest value measured for the optical density of the specific brain, the next curve includes the arithmetic means of the optical densities, and the third curve shows the highest value measured (Figure 6).

#### 2.4.1. Caudate-Putamen Complex

The graphs for the temporal development of the OD of the BoNT-A-treated CPu and the untreated CPu are shown in Figure 6. For the right CPu, a virtual maximum of the OD was registered 3.67 months after treatment. The OD of the contralateral CPu is clearly below that of the ipsilateral CPu (Figure 5 and Figure 6).

#### 2.4.2. Globus Pallidus

In the graphs calculated by cubic spine extrapolation, it can be seen that in the GP located in the BoNT-A-treated hemisphere, the OD initially dropped and reached a local minimum one month after treatment. Subsequently, the OD rose again to reach a local maximum at 3.67 months. In the contralateral GP, there was no formation of clear maxima within one year, but rather the OD dropped slightly over one year (Figure 6).

#### 2.4.3. Entopeduncular Nucleus

The mean OD of the EP on the BoNT-A-treated side rose briefly after the start of the experiment and remained in a plateau phase for months before steadily decreasing after about 6 months. On the contralateral side, the dynamics were similar but much weaker (Figure 6).

#### 2.4.4. Substantia Nigra

The right SN (ipsilateral to the injection side) of the BoNT-A-treated animals was the nucleus with the highest OD of all measured structures. The OD of the SN on the treated side decreased to a local minimum from 2 weeks after treatment to 1 month after treatment and then increased to a maximum at 3.67 months. After that, the OD steadily dropped again. The OD of the contralateral SN behaved similarly at a lower level and the maximum was reached after six months.

#### 2.4.5. Medial Thalamic Nuclei

The OD of the MThal on the treated side developed a maximum at six months to decrease thereafter. The contralateral nuclei behaved similarly at a lower level.

#### 2.4.6. Ventral Thalamic Nuclei

On the treated side, the OD of the VThal reached a slight maximum until approximately 2.48 months after treatment and then decreased. On the contralateral side, the OD hardly changed within one year.

#### 2.4.7. Motor Cortex

On the treated side, the OD initially decreased in the first weeks to reach a local minimum at three months, after which the OD increased again to a maximum at six months and then decreased again. On the contralateral side, the minimum was reached after one month, and the maximum was also at six months.

#### 2.4.8. Accumbens Nucleus

On the treated side, the OD dropped from an initially high value to a local minimum at 1 month and then rose to a maximum at 3.67 months. Thereafter, the OD steadily decreased again. On the contralateral side, the OD of the Acb increased to a maximum at six months and then decreased again.

#### 2.4.9. Pons

On both the treated and untreated sides, the OD decreased over time to a local minimum at three months post-treatment, then rose only slightly to a small maximum at six months, and then decreased again.

#### 2.4.10. Ventral Tegmental Area

The development of OD over time was similar for the VTA on the left and right sides. A local minimum was reached one month after treatment. The OD then increased to a local maximum at month four and only decreased thereafter.

#### 2.4.11. Habenular Nuclei

A slight local minimum was reached one month after treatment at both sides. The OD then increased to a local maximum at month six and decreased thereafter.

#### 2.4.12. Piriform Cortex

On the treated side, the OD dropped from a high value at two weeks after treatment to a local minimum one month after treatment, and then rose again to a local maximum at month three, then entered a plateau phase and dropped again after month nine.

On the untreated side, the OD of the Pir slightly increased at first and remained almost unchanged from one month to nine months after treatment, after which it slightly dropped.

#### 2.4.13. Basolateral Amygdala Nuclei

On the treated side, the OD of the BLAm initially showed a decrease up to one month after treatment and then increased again to a local maximum at three months after treatment. Subsequently, here, the OD decreased again. On the contralateral side, the first local minimum was reached at three months and the maximum at six months.

### 2.5. Comparison of Treated Animals with Control Animals

As there was only one sham group and this was analyzed 12 months post-injection, only possible significant differences between the sham group and the 12-month group are mentioned in the text and shown in Figure 5.

#### 2.5.1. Caudate-Putamen Complex

In a direct comparison of the right and left CPu of the sham animals (which survived 12 months) and the BoNT-A-treated animals (which survived 12 months), no significant differences could be detected. However, the Mann–Whitney U-test showed a clear trend towards a higher OD in the right CPu of BoNT-A-treated animals than in the right CPu of sham-treated animals even after 12 months (*p* = 0.056). Such a trend could not be detected in the left, untreated CPu (*p* = 0.548).

#### 2.5.2. Globus Pallidus

Twelve months after BoNT-A treatment, the OD of the GP on the injected side was significantly higher in the BoNT-A-treated animals than in sham animals. On the contralateral side, no significant differences in the OD of the GP of treated and sham-treated animals were registered (Figure 5b).

#### 2.5.3. Entopeduncular Nucleus

The OD of the EP of the BoNT-A-treated hemisphere (right) was significantly higher than the OD of the right EP in sham-treated animals after 12 months. On the contralateral side, there were no significant differences between BoNT-A-treated and sham animals after 12 months (Figure 5c).

#### 2.5.4. Substantia Nigra

The OD of the SN on the treated side of BoNT-A animals was significantly higher than the OD of the equilateral SN of sham animals after 12 months. No significant differences were detected in the contralateral SN between BoNT-A- and sham-treated animals (Figure 5d).

#### 2.5.5. Accumbens Nucleus

The differences of the OD of the Acb on the treated and untreated side of BoNT-A-treated animals to the OD of sham-treated animals were not significant after 12 months, but a trend was detected that the OD of the Acb on the treated side of BoNT-A animals was higher than that of sham-treated animals (*p* = 0.095; Figure 5h).

#### 2.5.6. Brain Regions with No Differences Compared to Sham-Treated Animals after 12 Months

For the following brain regions, no significant OD differences between BoNT-A-treated animals and sham-treated controls were detected for either the treated or the untreated sides at 12 months: MThal, VThal, MC, PN, VTA, Hb, Pir, and BLAm.

### 2.6. Correlation of the Optical Densities with the Distance from the Injection Site to the Gravitational Centers of the Examined Brain Areas

We examined the relation of the time of occurrence of the absolute maximum of the optical density in the respective brain area with the distance of its gravitational center towards the treated right CPu.

Using the stereotactic atlas data in neuroVIISAS, the distances between the gravitational centers of the studied regions and the gravitational center of the treated CPu were determined (Figure 7) [64,65,66] and shown in Table 2. Figure 8 shows the correlation between the measured maximum optical density of an investigated brain area and its distance from the injected CPu. Pearson’s correlation coefficient for this correlation is −0.55, which means there is a mean linear negative correlation between the level of maximal optical density of a brain area and its distance from the BoNT-A-treated CPu (*p* = 0.0039). This is indicated by the fact that we usually measured small maximum values of optical density in brain areas that were further away from the treated CPu.

Furthermore, we investigated the correlation of the time of occurrence of a maximum of the OD of a brain region and its distance to the injection site. Since the time points of the maxima in the Shapiro–Wilk test were not normally distributed, the rank correlation coefficient according to Spearman was used. We found a weak positive linear correlation between the distance of the respective gravitation centers and the time until the occurrence of a maximum, i.e., the further a region is away from the injection site, the later an increased immunoreactivity against cleaved SNAP-25 seems present. The rank correlation coefficient according to Spearman was 0.45 (*p* = 0.021). Accordingly, maxima of the mean OD occurred earlier the closer the region was to the injected CPu. Figure 9 shows the correlation of the time at which the maximum OD was reached as a function of the distance from the injected CPu.

In addition, it was examined whether further correlations exist between the mean optical densities of the investigated structures and their distance to the injected CPu. A mean linear relationship with distance from the injection site was determined, especially for the 3-month animals. The Pearson correlation coefficient was −0.58 (Figure 10).

### 2.7. Correlation of the Maximal Optical Density of the Examined Brain Areas with the Density of Their Connections to the Injection Site

The weighted efferent connection density from the injection site to the respective brain areas is shown in Table 2. These data were taken from neuroVIISAS and were obtained through literature searches for connectome research on the rat brain and were examined for statistical correlations with the OD. In the evaluation, only regions were included for which efferent connections of the CPu were described in the literature. So far, no efferent connections from the CPu with the Pn, the VThal, the Hb, the BLAm, the ACb and the Pir are known. A weak, non-significant positive correlation was found between the OD and the connection density to the injection site (rank correlation coefficient according to Spearman = 0.22; *p* = 0.367) (Figure 11).

### 2.8. Temporal Changes in the Ranking of the OD of the Brain Areas

For each time point of the study, we assigned a rank to the investigated brain areas based on their respective OD (Figure 12). Positions at the top of the ranking indicate a high OD, and positions at the bottom a low OD. Based on the determined OD ranks, the relative content of cleaved SNAP-25 of the examined brain regions was determined. The rank serves as an indirect indicator for the BoNT-A activity in the respective brain region. If structures change their position in the OD ranking between different time points, i.e., one region rises in the ranking within some time while another falls, then their mean OD has also changed. A change in the mean OD in turn means a change in the immunoreactivity for cleaved SNAP-25, i.e., the concentration of cleaved SNAP-25. Since the concentration of c-SNAP-25, as a direct cleavage product of BoNT-A, depends on the concentration of BoNT-A in the respective region, a change in the ranking position of a region indicates an upward transport of BoNT-A (in the case of an increase in the ranking) or a downward transport or reduction of BoNT-A (in the case of a decrease in the ranking). It was noticeable that in the course of one year, certain brain areas significantly changed their position in the ranking, i.e., they rose (Pn, MC, BLAm, Hb, VThal) or had fallen (EP, CPu, Acb, MThal, Pir) sharply, while only a few structures remained relatively constant (SN, GP, and VTA) in their position in the OD ranking. In fact, the three brain regions that showed the highest increase in the ranking in the course of the experiment (Hb +5 ranks, BLAm +5 ranks, Pn +4 ranks) also showed a large distance from the treated CPu (Hb = 4.0 mm, BLAm = 5.3 mm, and Pn, dP = 10.3 mm). Nuclei whose gravitational center is particularly close to the treated CPu (Acb = 3.1 mm, EP = 2.5 mm, MThal = 3.5 mm) showed a tendency to drop particularly strongly in their position of the OD rank (Acb ₋6 ranks, EP ₋4 ranks, MThal ₋3 ranks).

The SN located ipsilateral to the injection side always showed the highest OD at all time points despite its relatively large distance (dSN = 4.2 mm) from the BoNT-A-treated CPu. The connections of both nuclei with each other have been shown to be particularly pronounced.

## 3. Discussion

### 3.1. Cleaved SNAP-25 and Experimental Concept

We verified the success of BoNT-A treatment by detecting TH-positive and ChAT-positive BiVs in the treated CPu. BiVs are 1–5 µm swellings on catecholaminergic or cholinergic nerve fibers in the CPu that have been found in previous work exclusively after successful BoNT-A application into the CPu [30,31,63,67].

In previous experiments, we were able to demonstrate that an increased morbidity in comparison to sham-treated rats only begins with an injection of 5 ng of BoNT-A per CPu [33]. An application of 1 ng of BoNT-A did not lead to an increased mortality of experimental animals but had a clear effect on motor symptoms in the hemi-Parkinson model. The concentration of 1 ng of BoNT-A turned out to be the best working concentration and was also applied to the rats in this series of experiments. In line with previous results, no BoNT-A-associated increased morbidity was observed in this series of experiments, so that no animal dropped out during the course of the experiment.

The extraordinarily small amounts of BoNT-A injected into the CPu are difficult or impossible to visualize after further dilution due to its dispersion in the brain by diffusion or active transport and possible degradation [59,60,68]. For this reason, we visualized the enzymatic activity of BoNT-A by immunohistochemically labeling its cleavage product, cleaved-SNAP-25. Antonucci et al. [58] proved that the immunohistochemical detection of cleaved SNAP-25, generated by the catalytic activity of BoNT-A, was suitable for detecting the presence of BoNT-A in the brain parenchyma of rodents. The question may arise whether immunohistochemistry against cleaved SNAP-25 is the appropriate method to detect BoNT-A. It could be that the primary antibody binds to cleaved-SNAP-25, which was first formed in the respective part of the brain due to BoNT-A being transported to the corresponding area. However, it is also conceivable that cleaved SNAP-25 itself was transported from another brain region into the structure under investigation.Indeed, the OD, respectively the immunoreactivity for cleaved SNAP-25, appears to depend on the concentration of BoNT-A in the respective region and not only on an accumulation of cleaved SNAP-25. Thus, Foran et al. [69] found a t½ of >31 days for BoNT-A, whereas the half-life of BoNT-A-cleaved SNAP-25 was reported to be only 0.95 ± 0.2 days. BoNT-A thus appears to continuously cleave SNAP-25. The work of Holzmann et al. [32] already showed that cleaved SNAP-25 is also detectable in the ipsilateral motor and somatomotor cortex shortly after intrastriatal BoNT-A injection. Our results are consistent with this observation and additionally demonstrated the appearance of cleaved SNAP-25 in many other ipsilateral, and to a lesser extent, contralateral structures of the telencephalon, diencephalon, mesencephalon, and myelencephalon after intrastriatal BoNT-A injection.

We point out that with the present experimental design, we cannot prove a linear relationship between the detected cleaved SNAP-25 and the actually present catalytically active BoNT-A; however, an approximation about the presence of more or less BoNT-A seems to be realistic. We assume that BoNT-A does not passively diffuse into other brain areas after intrastriatal injection but reaches them through axonal transport. This is supported by the fact that cleaved SNAP-25 was mainly detected in brain areas that are connected with the CPu. A clear staining of the medial forebrain bundle also speaks for the transport of BoNT-A along fiber tracts (Figure 13). The medial forebrain bundle contains the nigrostriatal fiber tracts as well as connections from the VTA to the lateral hypothalamus and the Acb. Probably, BoNT-A cleaves SNAP-25 during its axonal transport in the fiber tracts, which is also about to be transported to the respective synapses [70]. The active transport of BoNT-A in the nervous system has also been described several times in the literature [58,59,60,68,71,72,73,74,75].

Since we were also able to detect cleaved-SNAP-25 in some areas of the brain on the contralateral side, e.g., in the olfactory bulb, it must be assumed that BoNT-A is also transported inter-hemispherically. Antonucci et al. [58] were able to demonstrate a similar phenomenon for the colliculi superiores and the respective contralateral retina. Akaike et al. and Cai et al. [71,76] were also able to detect cleaved SNAP in the contralateral spinal cord after peripheral injection. Connections of the olfactory bulb with the contralateral hemisphere are well-documented [77,78]. A sham-treated group of animals was examined 12 months after treatment. For animal welfare reasons, and to keep the number of required test animals as low as possible, for the other study time points, no sham groups were established.

### 3.2. Caudate-Putamen Complex

Interestingly, although the injection was made into the CPu, the maximum OD was never measured in the CPu, but in other brain areas. The reason for this could be that BoNT-A was already largely transported away from the CPu to other brain regions two weeks after injection, the earliest time point for measurement. This presumption is in line with the results of other research groups on intracerebral BoNT-A injections, which observed BoNT-A effects much earlier [58,79,80,81,82]. We deliberately decided to examine the brains of the experimental animals at the earliest after two weeks to make our results comparable with our earlier studies.

### 3.3. Olfactory Bulb

In the present work, a clear and intense immunoreactivity for BoNT-A-cleaved SNAP-25 was detected in the OB. Even nerve cell bodies were immunoreactive for cleaved SNAP-25. This was not the case in control animals. This result is remarkable in so far as no direct connections between CPu and the olfactory bulb have been described up to now. This implies that BoNT-A was transported transsynaptically. Transsynaptic transport of BoNT-A has already been reported by other research groups [58,80,81,82,83,84]. It is conceivable that BoNT-A reaches the OB indirectly, via anterograde transport across the corpus amygdaloideum [85], or that a retrograde transport from cortical structures (piriform cortex) occurs. Transsynaptic transport may occur in these brain structures, and from there to the OB further axonal transport takes place [86].

Significant immunoreactivity for cleaved SNAP-25 was measured in both the mitral cell layers of the OB and the Pir. Both structures play a crucial role in the processing of olfactory stimuli and the discrimination of different odors [87,88]. In the Pir, cholinergic signaling also plays an important role. It has been shown that disruption of cholinergic signaling leads to deficits in odor discrimination and perceptual learning [89,90]. It cannot be ruled out that an intrastriatal BoNT-A treatment influences the olfactory abilities by BoNT-A-mediated blockade of transmitter release in structures which are important for smelling. Our group demonstrated an improvement in the olfactory abilities of hemi-Parkinsonian rats after unilateral BoNT-A injection into the CPu [91].

### 3.4. Temporal Dynamics of the OD

The occurrence of OD maxima and thus of immunoreactivity to cleaved SNAP-25 probably depends on the distance to the CPu and speed of the axonal transport of BoNT-A. We assume that a greater distance to the CPu prolongs the transport time of BoNT-A, which then can only become catalytically active with a time delay in the respective brain area. This would lead to a later increase in the concentration of cleaved SNAP-25 and thus a later detectable higher OD, and respectively a higher immunoreactivity for cleaved SNAP-25. This is supported by our observation that the studied structures exhibit specific temporal dynamics with respect to their ranking for the highest mean OD. Two weeks, and respectively a few months, after BoNT-A injection, the OD is particularly high in brain regions located close to the injection site, and later, brain areas are at the top of the OD ranking that are located further away from the CPu.

Moreover, the connection density of a structure with the striatal injection site is also related to the preferential local enrichment of BoNT-A. Thus, a clear trend emerged that structures with high connection density to the CPu tended to be placed higher in the ranking of the maximum OD. This effect is particularly pronounced for the SN. For the SN, showing the highest OD, strong efferent and afferent connections with the CPu are described [64] (Figure 12 and Figure 14). Accordingly, for the first maximum OD to occur as early as possible and for the maximum OD to be as high as possible, the corresponding part of the brain must be located as close as possible to the injected CPu and have as high a density of connections with it as possible.

For some structures, we detected two local maxima of the OD, respectively, of the immunoreactivity to cleaved SNAP-25. At the first time point, two weeks after injection, relatively high values were measured, then the OD decreased to a local minimum about 1 month after treatment, and then increased to the next local maximum. Partly, a plateau phase followed, and then the OD decreased again (Figure 6g,i,w). Possibly, the OD already decreased again at the first monitoring time point. This would be in agreement with other groups, which were able to demonstrate central effects after BoNT-A injection much earlier, after one to seven days [58,79,80,81,82,83,84]. The presence of a second maximum may result from the fact that BoNT-A is transported on two alternative transport pathways or via two different circuits through one and the same brain region and is possibly transsynaptically transported several times. The measurements of the OD of the CPu also showed that the value was relatively low after two weeks (Figure 6, Table 1 and Appendix A), and the OD values increased shortly thereafter. This is noteworthy because the BoNT-A injection occurred in the CPu, and one would expect the OD to be particularly high there. Apparently, a first maximum of the catalytic activity of BoNT-A in the CPu was already exceeded within the first two weeks.

Another possible explanation for multiple extremata of OD is that BoNT-A is possibly transported or taken up into neurons by two different mechanisms. Using cultured hippocampal neurons, it was demonstrated that in addition to the known SV2C-mediated uptake of BoNT-A, another unknown mechanism for cellular uptake of BoNT-A must exist [79]. The different properties of the different brain areas regarding immunoreactivity towards BoNT-A-cleaved SNAP-25 may be explained by different densities of SV2C receptors.

### 3.5. Evaluation of Possible Consequences of Intrastriatal BoNT-A Injection

The dose of BoNT-A injected unilaterally into the CPu of the rats in this study exceeds the LD_50_ by a multiple, if the LD_50_ for BONT-A from mice is taken as a basis and extrapolated for rats [36]. To determine the LD_50_ of BoNTs, they are administered systemically to mice, usually by intraperitoneal injection. The systemic effect then causes death by respiratory paralysis. Nevertheless, all animals survived the treatment without any abnormalities being observed. We assume that because of the injection into the CNS, the blood–brain barrier prevented larger amounts of BoNT-A from reaching the periphery. In previous experiments, we were able to determine an increased body weight in mice after BoNT-A treatment. In mice and in rats injected with BoNT-A into the CPu twice within half a year, the CPu was reduced in size compared to the untreated CPu. A loss of neurons in the CPu after BoNT-A treatment could be excluded by stereologic analyses [31,63,67,92].

Idiopathic PD is a disease that develops over decades and probably starts in the peripheral nervous system and the olfactory system and then first shows neurodegenerative manifestations in the brainstem and increasingly spreads to the whole brain [93,94,95]. This leads not only to purely motor symptoms but also to vegetative, sensory, cognitive, and psychiatric disorders [1,2,3,19]. The experimental BoNT-A treatment in rats can never be a causal treatment but serves to experimentally attenuate the PD-associated hypercholinism. That anticholinergic therapy can improve motor and sensorimotor deficits in the Parkinsonian animal model has been shown in the past by Ztaou et al. [96].

The presence of cleaved SNAP-25 in different brain regions beyond the CPu proves that BoNT-A, after intrastriatal application, does not remain at its injection site. This makes it likely that intrastriatal injection of BoNT-A blocks not only acetylcholine release in the CPu, but also acetylcholine release and the release of other transmitters in additional brain regions.

It has been known for several decades that BoNT-A is able to block the release of other transmitters such as catecholamines, for example dopamine, from the presynapses, in addition to acetylcholine [97,98,99,100,101]. In previous work, we also demonstrated an effect of intrastriatal BoNT-A injection on catecholaminergic fibers, and we found dilations in catecholaminergic fibers, which we have termed “BiVs” [33,63,67]. We have also consistently measured the highest immunoreactivity for cleaved SNAP-25 in the SN, suggesting that BoNT-A has indeed been taken up by efferent axons of the SN in the CPu and may now be blocking the release of dopamine. Therefore, a theoretical risk of worsening Parkinson’s symptoms by intrastriatal BoNT-A injection, especially at the beginning of the clinically striking part of the disease, when about 30% of the dopaminergic afferents are still preserved, is conceivable.

As mentioned above, our results suggest that intrastriatal BoNT-A injection can also lead to an accumulation of BoNT-A in structures that are important for the sense of smell. In the 6-OHDA-induced hemi-Parkinson model of the rat, our research group could not detect any negative effects of intrastriatal application of BoNT-A on olfactory abilities of the animals. On the contrary, these properties were improved [91].

In the GP, among others, we measured a strong immunoreactivity. The GP contains a small population of cholinergic neurons. Evidence for a possible positive BoNT-A effect in the GP on motor functions in the hemi-Parkinson rat model has been shown [61,62].

We have also detected increased immunoreactivity for cleaved SNAP-25 in BLAm that play a role in anxiety-induced learning processes and exhibit hyperexcitability in anxiety and panic disorders [102,103]. Cholinergic and GABAergic modulations of the processing in the BLAm have been described [104,105,106]. This is in line with prior results of our group reporting reduced anxiety behavior after intrastriatal BoNT-A injection [32]. However, all these present and former results are not easily transferable to human Parkinson’s syndrome, as this is a bilateral disease that affects many other brain regions. BoNT-A may also behave differently in the human brain due to a different cytoarchitecture, different receptor concentrations (SV2 as a binding site for BoNT-A and the receptors of the different transmitter systems), and brain volumes that differ by dimensions.

Anticholinergic effects are not desired in brain regions other than the CPu. We were able to show that BoNT-A is transported to several other brain regions. Since we have detected increased immunoreactivity in the Pn after intrastriatal BoNT-A treatment, it seems conceivable that the pedunculopontine nucleus can also be influenced by an intrastriatal injection of BoNT-A. The pedunculopontine nucleus contains cholinergic neurons. Impairment of the PPN leads to motor deficits [22,107,108]. A possible influence on the PPN requires further histological investigations and motor tests by future experiments.

The nucleus basalis of Meynert, the hippocampus, and the cortex play important roles in cognitive abilities and bear cholinergic fibers, whereas we did not notice any increased immunoreactivity against cleaved SNAP-25 in the nucleus basalis of Meynert. We have already addressed the risk of a negative influence on cognitive abilities by intrastriatal BoNT-A injection in the past by means of various cognition tests. We did not find a clear negative influence of intrastriatal BoNT-A injection in rats on their cognitive abilities [32]. It must be noted that only a small part of cognitive abilities could be measured in the rats (spatial learning, working memory). Other cognitive abilities, as they are relevant in humans, e.g., the declarative memory, or the abilities to plan or to calculate, could not be assessed. In the same study, we found that a bilateral intrastriatal BoNT-A injection in healthy rats impairs balance and motor coordination. We suspect that the intervention by BoNT-A in a healthy system is responsible for this, possibly affecting several transmitter systems at the same time. Further experiments in Parkinsonian animal models are planned to verify or falsify this hypothesis.

### 3.6. Implications for Future Experiments

We suggest that the consistently and strikingly high immunoreactivity for cleaved SNAP-25 of the SN in BoNT-A-treated animals is due to the very high afferent and efferent connection density of the SN with the CPu [109]. As a consequence, the CPu, which is the largest nuclear area in the rat brain, probably transports BoNT-A through these many efferent and afferent fibers, converging anterogradely and retrogradely to the SN, which in turn is a much smaller nucleus, regarding the volume and number of neurons [110]. This could lead to the phenomenon that the concentration of BoNT-A is higher in the SN than in the treated CPu and explain why the OD in the SN is always higher than in the BoNT-A-treated CPu. In PD or in animal models of PD, a large part of the dopaminergic neurons of the SN that project with their axons into the CPu perish, so this transport pathway for BoNT-A into the SN should be omitted here. Ultimately, this assumption can only be proven by a new experiment. It is planned to repeat the present experiment in 6-OHDA hemilesioned rats. In these animals, at least all dopaminergic efferent fibers of the SN to the CPu are lost, so that these fibers for retrograde transport of BoNT-A would be omitted, and the immunoreactivity for cleaved SNAP-25 should be significantly weaker in these animals. The absence of enhanced immunoreactivity for cleaved SNAP-25 in the SN of these animals would confirm that dopaminergic fibers from the SN can transport BoNT-A into the SN via retrograde transport. Furthermore, immunofluorescence double staining against cleaved SNAP-25 as well as dopaminergic, glutamatergic, and GABAergic markers will be performed.

Since PD is a bilateral disease, the consequences of bilateral intrastriatal BoNT-A injections are being investigated in a genetic animal model of PD by extensive motor testing and histological examination [111].

## 4. Materials and Methods

### 4.1. Animals

Young adult male Wistar rats (*n* = 56) obtained from Charles River WIGA GmbH (Sulzfeld, Germany) were used for the studies. The weight of the rats at the time of stereotactic surgery was 300 g (±30 g). The rats had free access to water and food and lived in a temperature-controlled (22 ± 2 °C) room in which a light/dark rhythm of 12 h each was generated. Each experimental group initially consisted of 8 animals. All experiments were approved by the State Animal Research Committee of Mecklenburg-Western Pomerania (LALLF M-V 7221.3-1.1-003/13, 26 April 2013).

### 4.2. Stereotactic Injection of BoNT-A into the CPu

BoNT-A (Lot No. 13028A1A) was purchased by List (Campbell, CA, USA) via Quadratech( Surrey, UK). The injection solution was prepared by dissolving BoNT-A in phosphate buffered saline (PBS, pH 7.4) supplemented with 0.1% bovine serum albumin (BSA). The injection solution was freshly prepared from a frozen stock solution on the respective day of surgery. Rats were anesthetized with a mixture of ketamine (50 mg/kg, bela-pharm, Vechta, Germany) and xylazine (4 mg/kg, Rompun^®^, Bayer, Leverkusen, Germany). Injections of 1 ng of BoNT-A into the right CPu of rats were performed using a stereotactic frame (Kopf^®^, Tujunga, CA, USA). Sham-BoNT-A animals received only PBS + BSA. The BoNT-A solution was injected at two sites into the right CPu. The injection coordinates with reference to bregma were: anterior-posterior = +1.3 mm/−0.4 mm, lateral = −2.6 mm/−3.6 mm, and ventral = −5.5 mm/−5.5 mm [112,113]. A total of 2 µL of the BoNT-A solution or the vehicle solution was injected into the CPu, whereby 1 µL (0.5 ng of BoNT-A each) was injected in each of the consecutive coordinates. We used a complex protein-free form of BoNT-A. Accordingly, the amount of 1 ng of BoNT-A we used corresponds to approximately 227 international units [114,115].

### 4.3. Experimental Groups and Examination Times

For better comparability, the finalization times were strictly based on our previous experiments [33,57,63]. Seven experimental groups were established. Six groups were treated with 1 ng of BoNT-A in the right CPu, and one group received only the vehicle solution. Experimental groups were finalized after the following times after injection: 2 weeks, 1 month, 3 months, 6 months, 9 months, and 12 months. In addition, a group of control animals injected with the vehicle of BoNT-A was established and was finalized after 12 months.

### 4.4. Tissue Preparation and Histochemistry

Animals were sacrificed by an overdose of ketamine and xylazine and perfused with cold 0.9% saline and afterwards with cold 3.7% paraformaldehyde solution. The brains were removed, postfixed in 3.7% paraformaldehyde solution at 4 °C, cryoprotected overnight in 20% sucrose at 4 °C, frozen, and stored at −80 °C. Brains were cut in frontal sections (30 µm) on a cryostat (Leica Mikrosystems, Wetzlar, Germany) starting at the OB and stopping at the medulla oblongata. Brain slices were temporarily stored in cryoprotection solution at −20 °C until immunohistochemical staining. The slices of the left and right OB were stored separately.

Every 7th section was stained immunohistochemically for BoNT-A-cleaved SNAP-25 (primary antibody: mouse, 1:500, Emelca Bioscience, Escondido, CA, USA) and subsequent sections were Nissl-stained or stained for tyrosine hydroxylase (TH) or choline acetyltransferase (ChAT). Primary antibodies against tyrosine hydroxylase (monoclonal, cloneTH2, mouse, 1:1000, Sigma-Aldrich, St. Louis, MO, USA) and choline acetyltransferase (polyclonal, goat, 1:200, Millipore, Schalbach, Germany) were applied for staining of catecholaminergic and cholinergic structures. Afterwards, incubation of biotinylated horse anti-mouse IgG (Vector Laboratories, Burlingame, CA, USA, 1:67) or rabbit anti-goat IgG (Vector Laboratories, Burlingame, CA, USA, 1:67) occurred. The immunohistochemical labeling was visualized using a standardized avidin–biotin-peroxidase complex method (Vector Laboratories, Burlingame, CA, USA) and diaminobenzidine contrasted with ammonium nickel sulfate [63]. To avoid systemic errors that could be caused by possible slight differences in staining conditions, brains from different groups were stained simultaneously in each staining run.

### 4.5. Examined Brain Areas

The brain areas were examined by densitometry for a possible accumulation of BoNT-A in a time-dependent manner. For this purpose, the individual structures were examined for increased immunohistochemical labeling of cleaved SNAP-25. The following brain areas and fiber tracks were analyzed by densitometry both on the ipsilateral side and on the contralateral side:Caudate-Putamen complex (CPu)Globus pallidus (GP)Entopeduncular nucleus (EP)Substantia nigra (SN)Medial thalamic nuclei (MThal)Ventral thalamic nuclei (VThal)Motor cortex (MC)Accumbens nucleus (Acb)Pons (Pn)Ventral tegmental area (VTA)Habenular nuclei (Hb)Piriform cortex (Pir)Basolateral amygdala nuclei (BLAm)

In addition, specimens of the olfactory bulb (OB) were qualitatively examined from each time group.

### 4.6. Densitometry

The slides with the corresponding brain sections were scanned using the transmitted light method with a high-resolution scanner (Nexscan F4100, Heidelberger Druckmaschinen, Heidelberg, Germany). Digital 8-bit gray value images were generated with a resolution of 2650 dpi. Further computer-aided evaluation occurred using the program Icy 2.0.1.0 (BioImage Analysis unit Institut Pasteur Unite d’analyse d images quantitative, Paris, France). The nuclei, fiber tracts, and cortex areas to be examined were determined on the scans using the rat brain atlas of Paxinos and Watson [112] and were marked in the program ICY as respective ROIs with the functions “Area” or “Polygon”. Subsequently, the mean gray values of the ROIs were read out.

Furthermore, the gray values for the non-specific background of the respective slides had to be recorded. For this purpose, areas on the slide were marked on which no tissue was applied and defined as ROI for the unspecific background. The program Icy was used to determine the gray values of the regions of interest and the non-specific background. The mean gray values for each ROI were exported to an Excel spreadsheet (Excel 2016, Microsoft Corporation, Redmond, WA, USA). Subsequently, these gray values of the respective regions were used to calculate the optical density with the help of the program MATLAB^®^ (MATLAB R2020b, The MathWorks, Inc., Natick, MA, USA). For this purpose, the transmittance was first determined by dividing the gray value/the average intensity of the respective region by the gray value/the average intensity of the unspecific background (blank slide + coverslip): $T=\frac{I}{I_{0}}$.

Subsequently, the optical density was calculated by calculating the negative decadic logarithm of the transmission of the respective brain region, as: $E=-log_{10}T=-log_{10}\frac{I}{I_{0}}$.

High optical densities indicate stronger immunoreactivity and thus a higher concentration of cleaved SNAP-25.

### 4.7. Comparison and Correlation of the Obtained Data with Findings from Connectome Research Using the NeuroVIISAS

To correlate the data obtained on the occurrence of specific maxima of immune reactivity in different brain regions, we used the neuroVIISAS framework. NeuroVIISAS allows to perform extensive network simulations using information from publications that have collected tract-tracing data, and to retrieve and graphically display information from different rat brain areas about their connections and connection strengths with other brain areas [65,66]. NeuroVIISAS was used to obtain data on the distances between the gravitational centers of the injected CPu and the other investigated brain areas as well as the connectivity densities between the CPu and the investigated brain areas.

### 4.8. Statistics

The statistical program SPSS^®^ (IBM SPSS Statistics, Armonk, NY, USA) was used for statistical analysis. The Mann–Whitney U-Test was performed to compare the optical densities of the ipsilateral and contralateral hemispheres. Furthermore, the 12-month group was compared with the non-BoNT-A-treated sham group, and the groups (of different survival times) were compared with each other. The test for normal distribution was carried out with the Shapiro–Wilk test. Depending on whether a normal distribution was present, the correlation coefficient was calculated according to Pearson, otherwise the rank correlation coefficient was determined according to Spearman.

The minima, maxima, and mean values of the optical densities of each investigated structure at each time point were plotted as curves in graphs using Origin^®^ (OriginPro 2016G, OriginLab, Northampton, MA, USA) and its cubic-spline interpolation function. Correlation analyses were also performed using Origin^®^.

## 5. Conclusions

In previous experiments, BoNT-A was injected into the CPu of hemi-Parkinson rats to attenuate striatal hypercholinism in the dopamine-deficient CPu and improve motor abilities in hemi-Parkinson rats. We investigated whether and to what extent BoNT-A spreads to other brain areas after its injection into the CPu. BoNT-A spread to a number of brain areas in the telencephalon, diencephalon, mesencephalon, and rhombencephalon after injection into the right CPu of rats, and its areal concentration, studied as cleaved SNAP-25 immunoreactivity, depended on the distance to the injected CPu, and on its connection density to the CPu. Thus, there was a clear indication for axonal and transsynaptic transport of BoNT-A within the brain. Moreover, BoNT-A activity was still detectable in the brain one year after treatment. In summary, our results suggest that after injection into the CPu, BoNT-A is transported to many other brain areas and due to its properties, could block not only the exocytosis of acetylcholine but also of a number of other transmitters. This could result in a range of hypothetical impairments of sensory, motor, cognitive, and emotional regulation mechanisms. Whether and to what extent the demonstrated extensive BoNT-A transport manifests itself in actual disorders can only be determined in future behavioral experiments on healthy and bilateral Parkinson’s animal models.

## Figures and Tables

**Figure 1 ijms-24-01685-f001:**
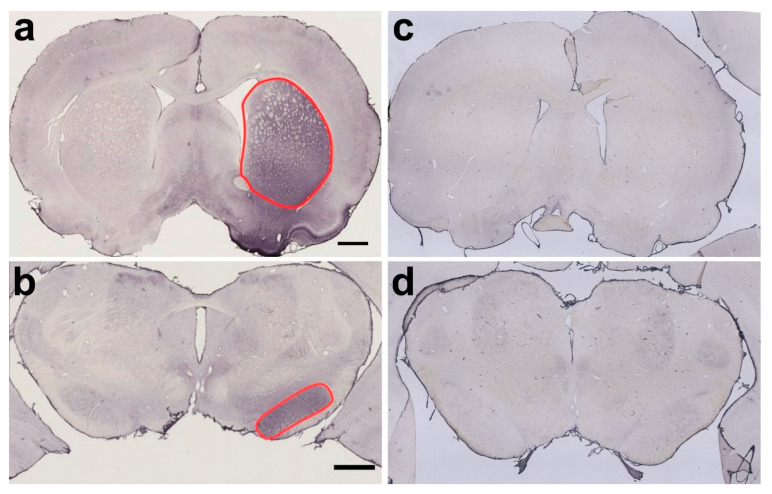
The figure depicts exemplarily 30 µm-thick frontal sections, immunohistochemically stained for cleaved SNAP-25, from a rat brain treated with 1 ng of BoNT-A on the right side one month (**a**,**b**) previously, and one rat brain treated with vehicle solution only 12 months previously (**c**,**d**). (**a**) A frontal section from the telencephalon of a BoNT-A-treated rat. The CPu of the right side is outlined in red. (**b**) A frontal section from the midbrain of the same animal. The SNpc is marked in red. (**c**) The frontal section of the telencephalon and (**d**) of the midbrain of a vehicle-treated rat. Note the differences in the strength of the staining in the CPu and the SNpc. The scale bar in (**a**) corresponds to 1 mm in (**a**,**c**). The scale bar in (**b**) corresponds to 1 mm in (**b**,**d**).

**Figure 2 ijms-24-01685-f002:**
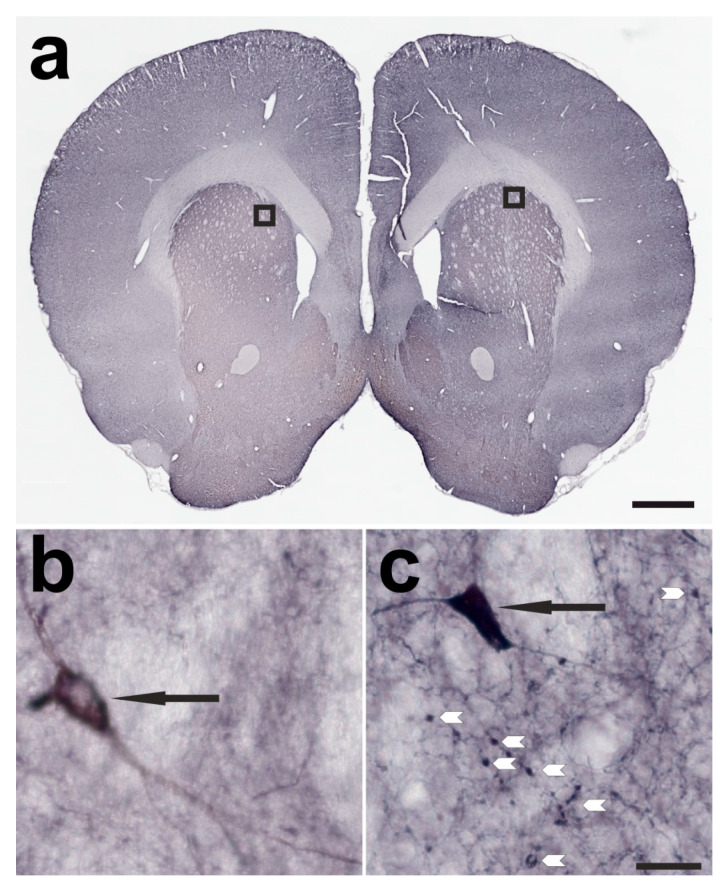
A frontal section through the telencephalon of a rat treated with 1 ng of BoNT-A in the right CPu one month earlier. The specimen was immunohistochemically stained against ChAT. In (**a**), an overview of the frontal section is presented. In (**b**), the marked box of the left (untreated) CPu is enlarged. A cholinergic interneuron is marked with a black arrow. In (**c**), a comparable area of the right (BoNT-A-treated) CPu is enlarged, also depicting a cholinergic interneuron, marked with a black arrow. Additionally, ChAT-positive BiVs (BoNT-A-induced varicosities) are marked with white arrowheads. The scale bar in (**a**) corresponds to 1 mm, the bar in (**c**) corresponds to 20 µm and is valid for (**b**,**c**).

**Figure 3 ijms-24-01685-f003:**
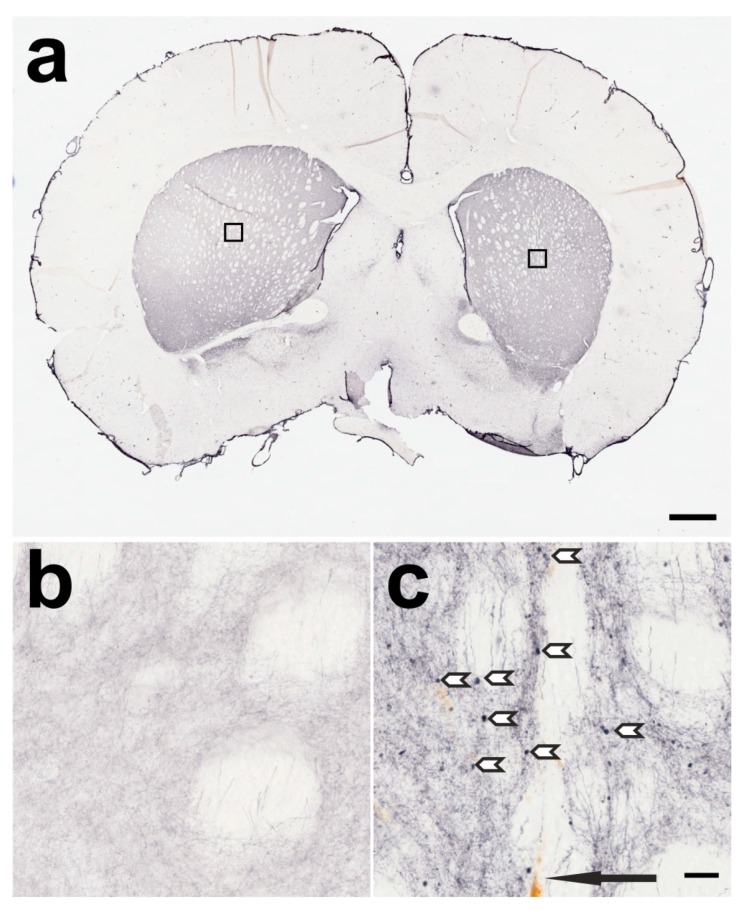
A frontal section through the telencephalon of a rat treated with 1 ng BoNT-A in the right CPu one month earlier, immunohistochemically stained against TH. (**a**) An overview of the frontal section, and (**b**) the enlarged area of the left (untreated) CPu marked with a box above on the left side in (**a**). In (**c**), a comparable area of the right (BoNT-A-treated) CPu is enlarged. TH-positive BiVs (BoNT-A-induced varicosities) are marked with white arrowheads. The black arrow points to the injection channel with brownish colored macrophages. The scale bar in (**a**) corresponds to 1 mm, the bar in (**c**) corresponds to 20 µm and is valid for (**b**,**c**).

**Figure 4 ijms-24-01685-f004:**
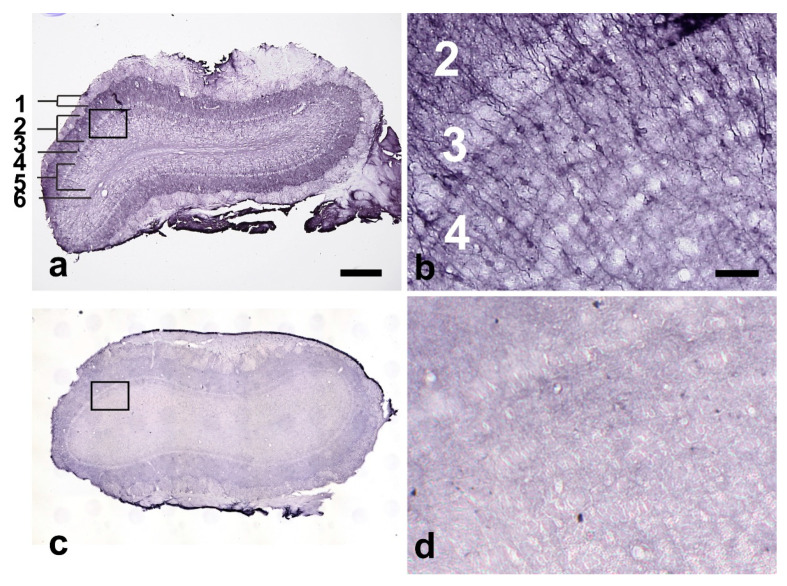
The figure shows 30 µm-thick sections immunohistochemically stained for cleaved SNAP-25 of the right olfactory bulb of a rat injected with 1 ng of BoNT-A into the right CPu 2 weeks before (**a**,**b**) and a rat injected with vehicle solution only 12 months before (**c**,**d**). (**a**,**c**) Overview of the olfactory bulb section. The rectangle marked in (**a**) is shown enlarged in (**b**). The rectangle marked in (**c**) is shown enlarged in (**d**). Cleaved SNAP-25-positive perikarya and fibers are detectable in (**a**,**b**) but not in (**c**,**d**). Note that in addition to the nerve fibers, neuronal cell bodies are clearly immunoreactive for cleaved SNAP-25 in (**a**,**b**). The individual layers of the olfactory bulb are annotated by numbers (**a**,**b**) and markings (in (**a**)), 1: periglomerular layer, 2: external plexiform layer, 3: mitral cell layer, 4: internal plexiform layer, and 5: granular cell layer. The scale bar in (**a**) corresponds to 500 µm for (**a**,**c**), and that in (**b**) to 50 µm for (**b**,**d**).

**Figure 5 ijms-24-01685-f005:**
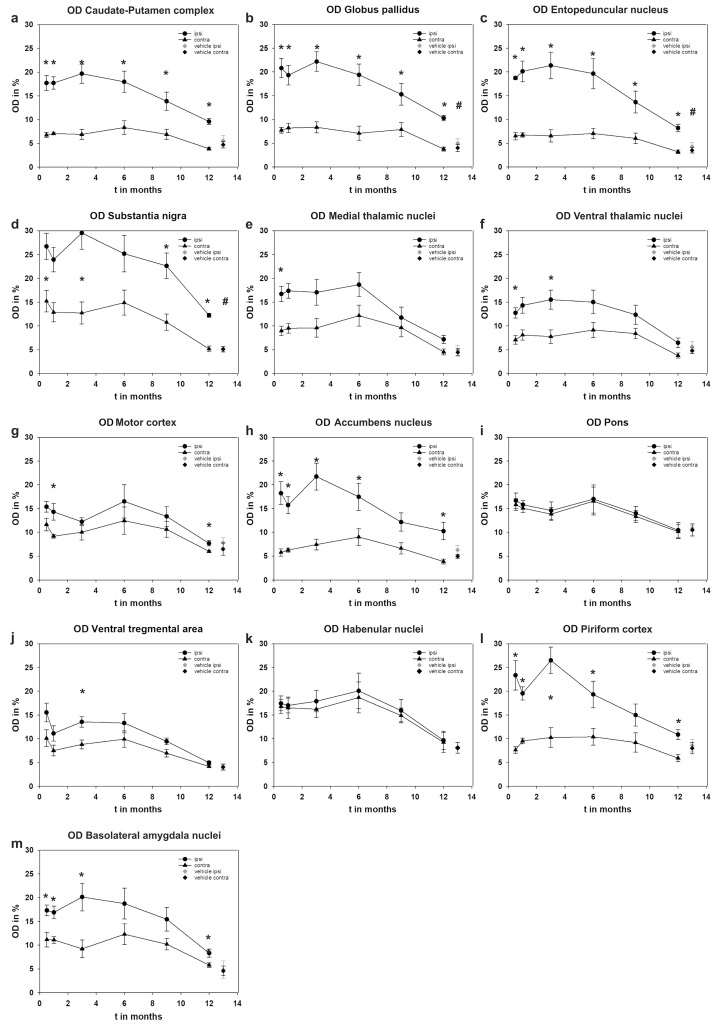
Time courses of areal-specific mean optical densities (OD) of cleaved SNAP-25. The *x*-axis shows the survival time after BoNT-A, and the *y*-axis shows the OD in %. The mean OD of the treated side (black circles) and the untreated sides (black triangles) are plotted ± the standard error of the mean. For comparison, the OD of the sham-injected hemispheres 12 months post-injection are marked with gray diamonds, and the values of the respective contralateral sides with a black diamond. For clarity, these values were plotted at month 13. At each time point, the OD of the ipsilateral side and contralateral sides were compared using the Mann–Whitney U-test. Significant differences between the hemispheres are marked with asterisks (*). If there were significant differences between the right (treated) hemisphere and the right hemisphere of sham animals at 12 months after treatment, a diamond (#) is shown above the values of the sham animals. In subfigures (**a–m**), one can see the development of mean OD for the following brain structures on both the contralateral and ipsilateral sides to the injection side: (**a**) Caudate-Putamen complex, (**b**) Globus pallidus, (**c**) Entopeduncular nucleus, (**d**) Substantia nigra, (**e**) Medial thalamic nuclei, (**f**) Ventral thalamic nuclei, (**g**) Motor cortex, (**h**) Accumbens nucleus, (**i**) Pons, (**j**) Ventral tegmental area, (**k**) Habenular nuclei, (**l**) Piriform cortex, (**m**) Basolateral amygdala nuclei.

**Figure 6 ijms-24-01685-f006:**
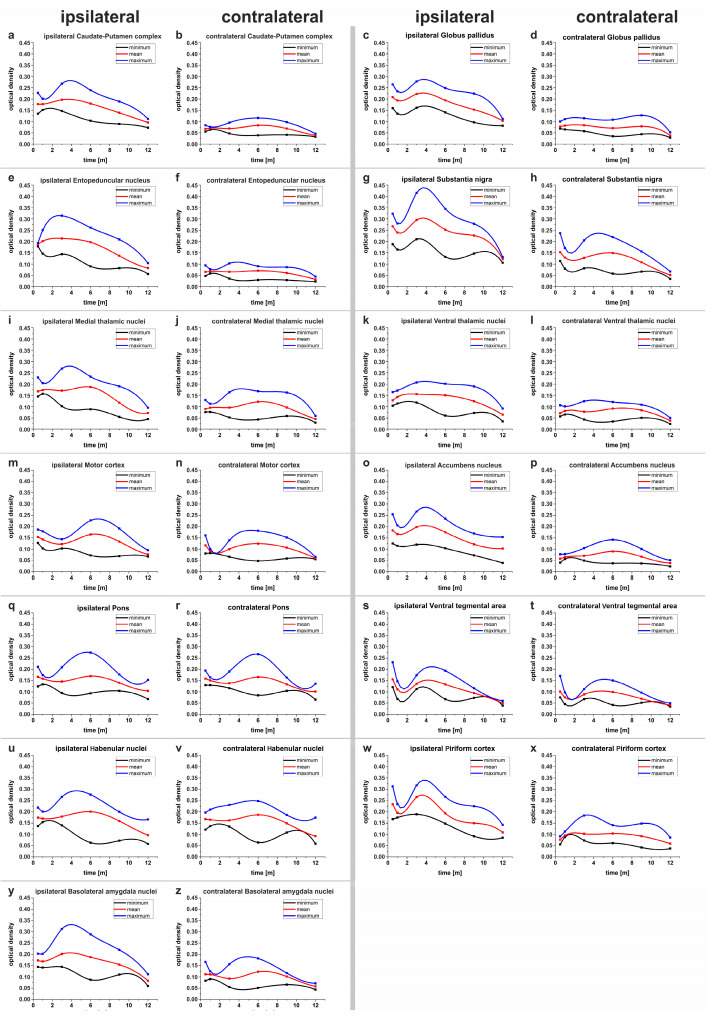
The diagrams show the course of the ODs over time in the course of the experiment for the investigated nuclei or brain areas of each side. Three curves are plotted in each diagram. The red curve shows the course of the mean optical density. The blue curve shows the course of the measured maximum values of the OD at one point in time. The black curve shows the course of the measured minimum values of the OD at one point in time. The curves were generated from the measured values by cubic-spine interpolation using the program Origin^®^. The diagrams for the brain areas ipsilateral to the BoNT-A treatment are shown in the first and third columns, and the corresponding diagrams for the contralateral (untreated) side in the second and fourth columns. The extrapolated curves for the development of minimum measured OD, mean OD, and maximum measured OD are shown for brain parts ipsilateral to the injection site and contralateral to the injection site in subfigures (**a**–**z**): (**a**) ipsilateral Caudate-Putamen complex (**b**) contralateral Caudate-Putamen complex (**c**) ipsilateral Globus pallidus (**d**) contralateral Globus pallidus (**e**) ipsilateral Entopeduncular nucleus (**f**) contralateral Entopeduncular nucleus (**g**) ipsilateral Substantia nigra (**h**) contralateral Substantia nigra (**i**) ipsilateral Medial thalamic nuclei, (**j**) contralateral Medial thalamic nuclei, (**k**) ipsilateral Ventral thalamic nuclei (**l**) contralateral Ventral thalamic nuclei (**m**) ipsilateral Motor cortex (**n**) contralateral Motor cortex (**o**) ipsilateral Accumbens nucleus (**p**) contralateral Accumbens nucleus (**q**) ipsilateral Pons (**r**) contralateral Pons (**s**) ipsilateral Ventral tegmental area (**t**) contralateral Ventral tegmental area (**u**) ipsilateral Habenular nuclei (**v**) contralateral Habenular nuclei (**w**) ipsilateral Piriform cortex (**x**) contralateral Piriform cortex (**y**) ipsilateral Basolateral amygdala nuclei (**z**) contralateral Basolateral amygdala nuclei.

**Figure 7 ijms-24-01685-f007:**
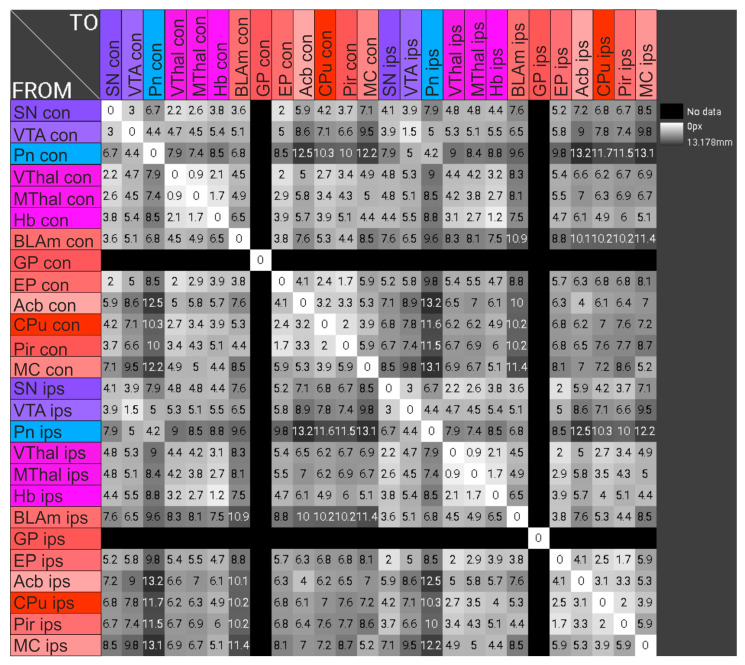
The distances of the gravitational centers of selected brain areas to each other, in mm, were determined with the program neuroVIISAS. This again draws on datasets from a large number of publications on connectome studies. With the help of neuroVIISAS, the distances of the gravitational centers of the examined brain areas to the injected CPu were determined.

**Figure 8 ijms-24-01685-f008:**
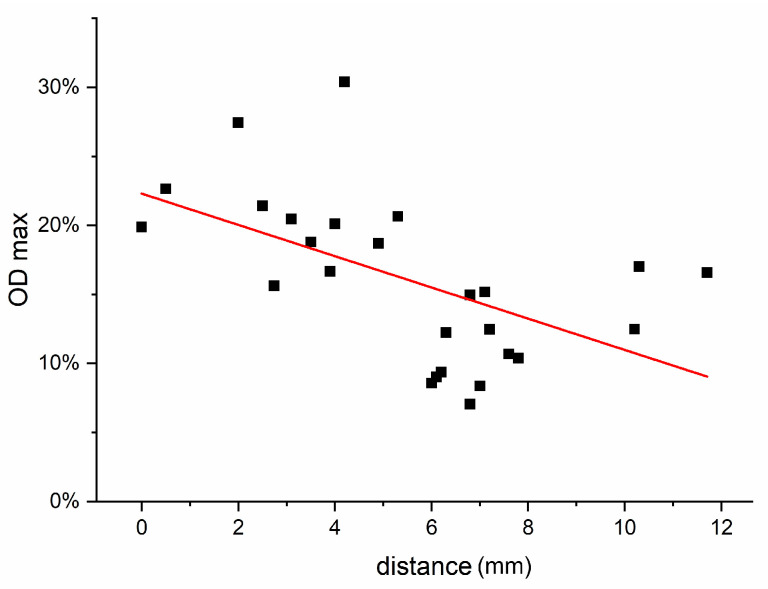
Graphic presentation of the maximum mean optical densities of all ipsilateral and contralateral examined brain areas with the distance (mm) to the treated CPu. A mean negative linear correlation (Pearson correlation coefficient = −0.55; *p* = 0.0039) was found between the distance to the BoNT-A-treated CPu and the maximum values of the interpolated curves for OD. The red line is the regression line calculated from the compared values.

**Figure 9 ijms-24-01685-f009:**
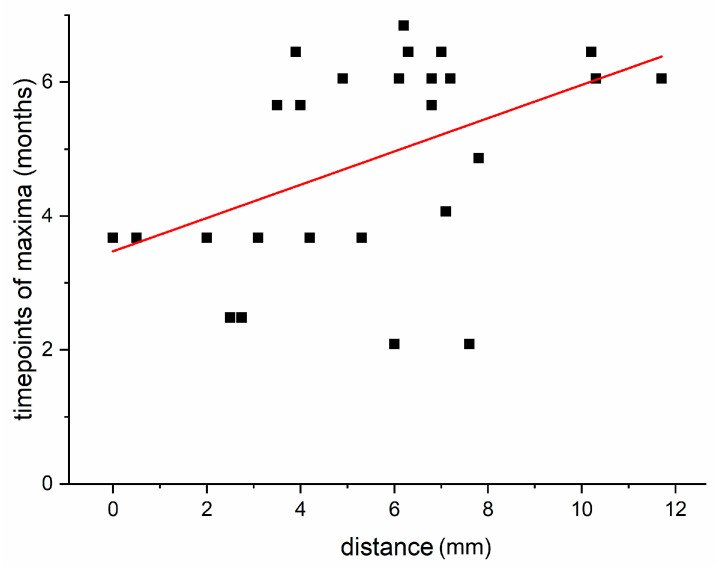
Graphic presentation of the time points of maximum mean optical density of all brain areas with their distance (mm) from the ipsilateral CPu. Plotted are the values for the temporal occurrence of the virtual maximum OD of the examined brain areas as well as the distance of their gravitational centers to that of the BoNT-A-injected CPu. There is a slight linear correlation (rank correlation coefficient according to Spearman = 0.45; *p* = 0.021) of the time point at which the maximum OD occurs (in the interpolated curves) and the distance from the injection site. The red line is the regression line calculated from the compared values.

**Figure 10 ijms-24-01685-f010:**
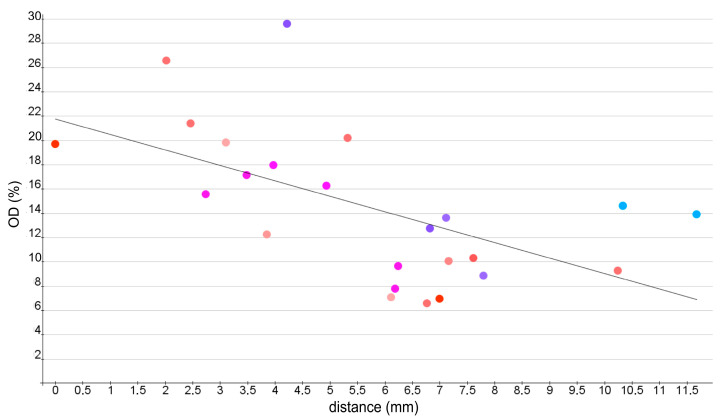
On the diagram, the OD of the examined brain areas is plotted against their distance from the treated CPu. The values of an ipsilateral and corresponding contralateral brain area are plotted with the same color. The black line is the regression line.

**Figure 11 ijms-24-01685-f011:**
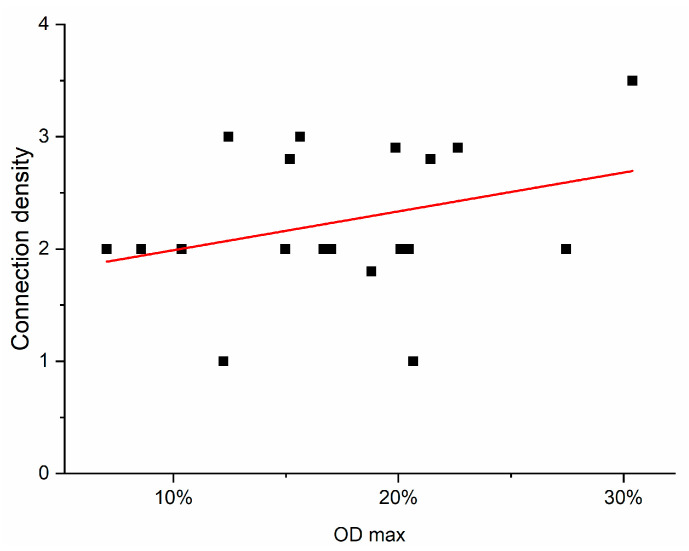
Correlation of the virtual maximum OD determined by cubic-spine interpolation and the connection density of the examined brain structures with the BoNT-A-treated CPu. The values come from animals finalized after 3 months. The connection density was taken from neuroVIISAS. A mean linear relationship with distance from the injection site was found (correlation coefficient is 0.22). The red line is the regression line calculated from the compared values.

**Figure 12 ijms-24-01685-f012:**
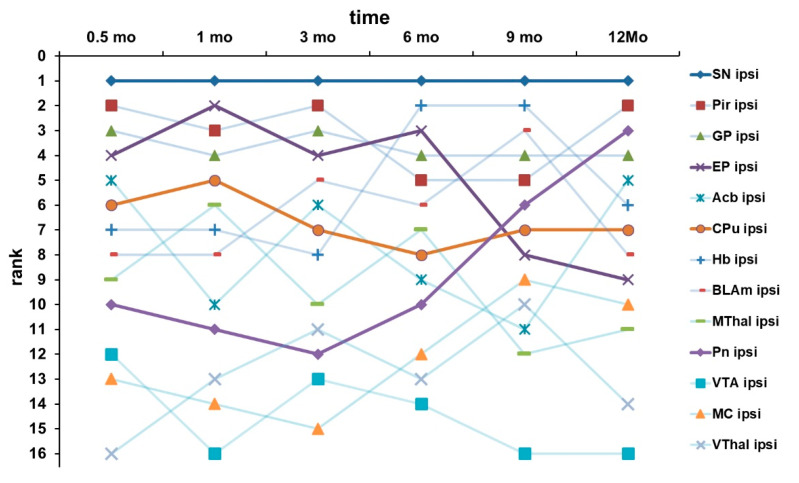
The diagram shows the ranking in terms of OD of selected brain areas examined in the BoNT-A-treated hemisphere after immunohistochemical staining of cleaved SNAP-25. The ranks of the respective structures are plotted on the *y*-axis, and on the *x*-axis are the time points of the examinations. It can be seen that the SN always has the highest immunoreactivity. For some structures, the ranking does not change or hardly changes, for other structures the ranking decreases over time, and for still other structures, the ranking increases over time. To make this more visible, the graphs of most structures were drawn transparently, and only the curve for the SN and the CPu as representatives of structures whose rank remains relatively constant, the graph of the EP as representative of a structure whose rank decreases over time, and the graph of the Pn as representative of a structure whose rank increases over the course of a year were drawn more opaque and thicker. All structures originate from the treated hemisphere and for this reason have the addition “ipsi” for ipsilateral. SN—substantia nigra; Pir—piriform cortex; GP—globus pallidus; EP—entopeduncular nucleus; Acb—Accumbens nucleus; CPu—Caudate-Putamen complex; Hb—nuclei habenulares; BLAm—basolateral amygdala nuclei; MThal—medial thalamic nuclei; VTA—ventral tegmental area; MC—motor cortex; VThal—ventral thalamic nuclei.

**Figure 13 ijms-24-01685-f013:**
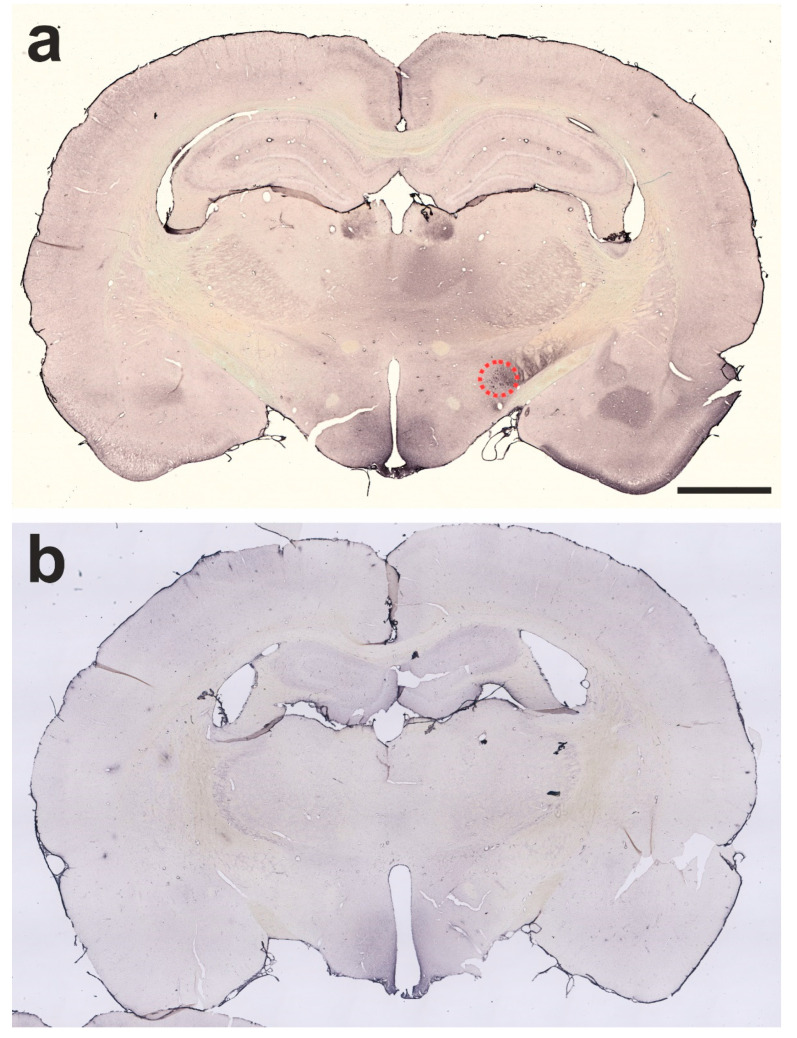
The 30 µm-thick frontal sections immunohistochemically stained for cleaved SNAP-25 of the diencephalon of a rat treated with 1 ng of BoNT-A on the right side one month previously (**a**) and a rat treated with vehicle solution 12 months previously (**b**). The medial right forebrain bundle is marked by a red dash line (**a**). A high immunoreactivity of the right medial forebrain bundle is clearly recognizable in (**a**). The scale bar corresponds to 2 mm for (**a**,**b**).

**Figure 14 ijms-24-01685-f014:**
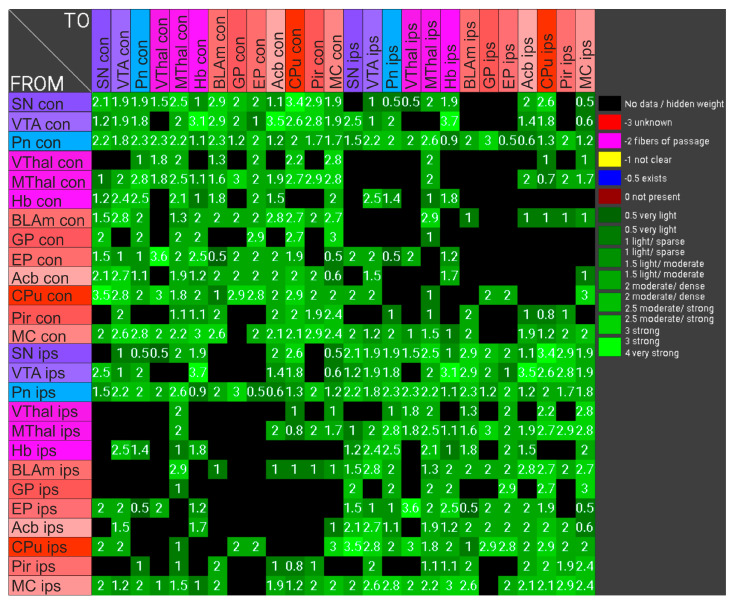
The connection frequencies between the examined brain parts, obtained from the program neuroVIISAS, were weighted, and weighting was plotted between ₋3 and 4 and color-coded. The information from neuroVIISAS is based on extensive literature data. Weighting: ₋3 unknown; ₋2 fibers of passage; ₋1 not clear; ₋0.5 exists; 0 not present; 0.5 very light; 1 > 0.5 light/sparse; 1.5 > 1 light/moderate; 2 > 1.5 moderate/dense; 2.5 > 2 moderate/strong; 3 > 2.5 strong; 4 very strong.

**Table 1 ijms-24-01685-t001:** The table summarizes the mean optical densities in % of the structures studied for all experimental animal groups: 100% stands for complete opacity and 0% for complete light transmission of the specimen. For clarity, the table has been divided into four parts. The addition “ips” to the column name stands for ipsilateral (region to the injection side) and the addition “con” for contralateral (to the injection side). CPu: caudate-putamen complex, GP: globus pallidus, EP: entopeduncular nucleus, SN: substantia nigra, MThal: medial thalamic nuclei, VThal: ventral thalamic nuclei, MC: motor cortex, Acb: accumbens nucleus, Pn: pons, VTA: ventral tegmental area, Hb: habenular nuclei, Pir: piriform cortex, BLAm: basolateral amygdala nuclei. See also Appendix A.

Group	CPu ips	CPu con	GP ips	GP con	EP ips	EP con
**2w**	17.713	6.750	20.814	7.734	18.725	6.512
**1m**	17.708	7.061	19.325	8.212	20.130	6.730
**3m**	19.672	6.897	22.187	8.357	21.359	6.554
**6m**	17.964	8.327	19.395	7.092	19.657	7.039
**9m**	13.875	6.893	15.297	7.867	13.649	6.020
**12m**	9.589	3.857	10.286	3.770	8.195	3.193
**group**	**SN ips**	**SN con**	**MThal ips**	**MThal con**	**VThal ips**	**VThal con**
**2w**	26.734	15.226	16.732	8.965	12.740	7.057
**1m**	23.953	12.880	17.387	9.512	14.319	8.090
**3m**	29.532	12.723	17.072	9.580	15.531	7.750
**6m**	25.204	14.904	18.693	12.166	15.027	9.129
**9m**	22.641	10.771	11.743	9.621	12.350	8.386
**12m**	12.235	5.174	7.156	4.519	6.449	3.764
**group**	**MC ips**	**MC con**	**Acb ips**	**Acb con**	**Pn ips**	**Pn con**
**2w**	15.391	11.646	18.234	5.767	16.666	15.841
**1m**	14.310	9.175	16.651	6.432	15.819	15.055
**3m**	12.223	10.011	19.791	7.041	14.573	13.842
**6m**	16.504	12.446	17.469	9.007	16.997	16.568
**9m**	13.338	10.615	12.145	6.628	13.988	13.337
**12m**	7.674	5.992	10.244	3.852	10.425	10.134
**group**	**VTA ips**	**VTA con**	**Hb ips**	**Hb con**	**Pir ips**	**Pir con**
**2w**	15.534	10.107	17.441	16.774	23.368	7.645
**1m**	11.112	7.518	17.012	16.520	19.556	9.526
**3m**	13.558	8.811	17.899	16.228	26.510	10.257
**6m**	13.302	9.911	20.092	18.696	19.335	10.399
**9m**	9.453	6.965	15.935	14.901	14.983	9.204
**12m**	4.969	4.170	9.655	9.238	10.880	5.923
**group**	**BLAm ips**	**BLAm con**				
**2w**	17.311	11.152				
**1m**	16.893	11.099				
**3m**	20.142	9.220				
**6m**	18.744	12.288				
**9m**	15.441	10.180				
**12m**	8.299	5.799				

**Table 2 ijms-24-01685-t002:** The table lists the virtual maximal OD, the time points for the occurrence of the virtual maximal OD, the distances of their gravitational centers with that of the treated CPu, and the density of connections with the treated CPu for all brain areas studied. The lines with a gray background contain the values of the brain areas from the contralateral hemisphere. The distances and the connection densities were taken from the program neuroVIISAS.

Structure	Interpolated Maximal OD (%)	Interpolated Time Point of Maximal OD	Distance to Treated CPu (mm)	Density of Connections to Treated CPu	Density of Connections in Words
**BLAm**	20.662	3.67241	5.3	1	light/low
**MC**	16.669	6.44828	3.9	2	moderate/dense
**GP**	22.65	3.67241	0.5	2.9	strong
**Acb**	20.461	3.67241	3.1	2	moderate/dense
**EP**	21.422	2.48276	2.5	2.8	strong
**Hb**	20.105	5.65517	4	2	moderate/dense
**Pir**	27.446	3.67241	2	2	moderate/dense
**Pn**	17.005	6.05172	10.3	2	moderate/dense
**SN**	30.397	3.67241	4.2	3.5	very strong
**CPu**	19.872	3.67241	0	2.9	strong
**MThal**	18.801	5.65517	3.5	1.8	moderate/dense
**VTA**	15.18	4.06897	7.1	2.8	strong
**VThal**	15.624	2.48276	2.745	3	strong
**BLAm**	12.475	6.44828	10.2	-	not present
**MC**	12.444	6.05172	7.2	3	strong
**GP**	8.569	2.08621	6	2	moderate/dense
**Acb**	9.012	6.05172	6.1	-	not present
**EP**	7.041	6.05172	6.8	2	moderate/dense
**Hb**	18.697	6.05172	4.9	-	not present
**Pir**	10.402	6.05172	7.6	-	not present
**Pn**	16.577	6.05172	11.7	-	not present
**SN**	14.967	5.65517	6.8	2	strong
**CPu**	8.368	6.44828	7	-	not present
**MThal**	12.223	6.44828	6.3	1	light/low
**VTA**	10.365	4.86207	7.8	2	moderate/dense
**VThal**	9.357	6.84483	6.2	-	not present

## Data Availability

Not applicable.

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
