# Peer review of "Distribution of Cleaved SNAP-25 in the Rat Brain, following Unilateral Injection of Botulinum Neurotoxin-A into the Striatum"

_ijms, 2023, doi:10.3390/ijms24021685_

Round 1

Reviewer 1 Report

In this study authors conducted a meticulous histological analysis of the brain after botulinum toxin A administration in the striatum to conclude about the extension of the toxin effects, ultimately envisioning it's use for the treatment of hypercholinism in Parkinson's disease. The reserach topic is relevant and the manuscript is clearly written, however there are some matters that need further attention.

- This work was exclusively done in healthy Wistar rats and no Parkinson's model was used. How do changes in histological features and innervation of the pathological brain can affect the effects of the toxin? It is important to validate the findings in the context of parkinsonism.

- Authors stated in the methods section that "Each experimental group initially consisted of 8 animals". Did animals die in the timecourse of the experimental protocol due to the injection of the botulinum toxin in the striatum?

- The extent of the toxin effect was evaluated in a number of brain regions after a single injection in the striatum. However, the spread of the toxin solution and the extent of SNAP-25 cleavage will depend on the dose and volume administered. What was the total volume injected? How does the dose of 1ng correlates with clinicaly relevant doses in units?

- Consider adding images of the control groups to clearly show immunorecativity against cleaved SNAP-25, ChAT and TH and evidence what was considered immunolabeling and backgound.

Author Response

Please find our answers in the attached Word file.

Kind reagrds

Alexander Hawlitschka

Reviewer 2 Report

Major concerns

1.       The motivation that it would be possible to treat hemiparkinsonian rats with botox A injections need better explanation. It comes out as a rather bizarre suggestion. What is the need for treating hemiparkinsonian rats? (Just refrain from lesioning…) The study would benefit from a somewhat different frame, whether that be focussing on the botulinum toxin transport or providing some indication that SNAP-25 cleaving as such could be a clinically feasible approach for treating neurodegenerative parkinsonism.

2.       Uptake of BoNTA in TH-positive cells must be a concern. This is not discussed.

3.       The rationale behind BoNTA injection into the striatum is supposed to be treatment of hypercholinergically induced parkinsonism. The fact that the spread of BoNTA effects are not determined in dopamine denervated animals should be discussed, in particular as the highest OD was always present in SN. Alternatively it would be useful to know it the OD is present in GABA-ergic, glutamatergic or dopaminergic structures in the SN.

4.       The finding that CPu never has the highest OD is contrary to the preformed hypothesis that BoNTA could be clinically useful as a method to induce local anticholinergic effect in the striatum without widespread negative effects on the important cholinergic CNS transmission. This unexpected outcome is hardly mentioned.

5.       The sham animals were only evaluated after a year. It is difficult to interpret the contralateral SNAP25 OD during the preceding 12 months without knowing what the background signal is in sham injected animals during this time.

 Minor

L33 – not only one of, but the most frequent neurodegenerative movement disorder (unless ET is viewed as neurodegenerative which is rightfully disputed)

L40 – “ partly responsible for a large part“ is unclear wording. It is not generally accepted that striatal hypercholinism is very important. Eg. acetylcholinesterase inhibitors have little or no effect on motor function in dopamine substituted PD patients and the efficacy of anticholinergics is low in most individuals with PD. This does of course not exclude that it is important in the 6OHDA-hemilesioned rat, but that is after all a model for PD pathophysiology.

The current introduction is in this regard not well balanced in my opinion.

Figure 12 is very difficult to interpret and I am not convinced about its value.

The section on valuation of therapeutic potential does not discuss the practicalities of injecting BoNTA into the human brain for therapeutic purpose.

Author Response

Please find our answers in the attached Word file.

Reviewer 3 Report

In the manuscript entitled ‘Distribution of cleaved SNAP-25 in the rat brain, following unilateral injection of Botulinum Neurotoxin-A into the striatum’, the authors systematically measured the kinetics of Botox transport after being unilaterally injected into striatum of rats. Generally it was a fine designed work with explicit description, but I am concerning the significance would be more limited in the anatomical evidence across brain regions. Whether will the stereotactic injection of Botox play a role in PD rats needs further validating. Overall, I suggest a minor revision.

Minor

1. The present work was performed on healthy rats. But in PD animals, TH positive cells in stratum might have degenerated, whether Botox transport in a similar mode is questionable.

2. The amount of Botox A used was about 227 international units, a very high dose. Why not any degenerative change or behavior aberrance was observed? (line 210)

3. The authors used rats injected with vehicle solution before 12 months but not before 1 month as controls (Figure 1). Please explain the reason.

4. What did BiVs mean (Figure 2, Figure 3)?

5. Error bars and n numbers should be provided in Table 1.

6. The dysregulation of neurotransmission inside the SN and stratum was complex in PD. That means the usage of Botox triggering the single inhibitory effect would be not easily applicable. 

Author Response

Dear Reviewer,
Please find the answers to your points in the attached Word file.

Yours sincerely

Alexander Hawlitschka

Round 2

Reviewer 1 Report

Authors improved the manuscript in accordance with reviewers comments. The work is now clearly described and limitations and future works were described. 

Author Response

Dear Reviewer,

on behalf of all the authors, I would like to thank you for taking the time to review our manuscript.

Kind regards

Alexander Hawlitschka

Reviewer 2 Report

My main concerns with this work have not been addressed. The authors are doing their work a disservice by emphasizing clinical applicability. This is not near the suggestion of clinical application. The basic assumption that it is possible to locally inhibit cholinergic striatal activity is demonstrated to be incorrect and the consequence of this is neglected. PD is a bilateral disease. The authors have previously demonstrated negative effects of bilateral injections. The context of this work should be limited to understanding the effect and spread of Botox A in the brain. 

What the authors are now sort of suggesting is that PD patients have tubes implanted to regularly inject their brains with neurotoxin that spreads throughout the brain. This is a grave violation of the non nocere principle.

Author Response

Dear Reviewer,

on behalf of all the authors, I would like to thank you for taking the time to review our manuscript.

Please find our answers and changes in the attached Word file.

With kind regards

Alexander Hawlitschka
